# Transcriptome Analysis Reveals the Mechanism by Which Exogenous Melatonin Treatment Delays Leaf Senescence of Postharvest Chinese Kale (*Brassica oleracea* var. *alboglabra*)

**DOI:** 10.3390/ijms25042250

**Published:** 2024-02-13

**Authors:** Hongmei Di, Chenlu Zhang, Aolian Zhou, Huanhuan Huang, Yi Tang, Huanxiu Li, Zhi Huang, Fen Zhang, Bo Sun

**Affiliations:** College of Horticulture, Sichuan Agricultural University, Chengdu 611130, China; 2023105007@stu.sicau.edu.cn (H.D.); 2022205036@stu.sicau.edu.cn (C.Z.); 2022305050@stu.sicau.edu.cn (A.Z.); hh820423@163.com (H.H.); 13920@sicau.edu.cn (Y.T.); 10650@sicau.edu.cn (H.L.); huangzhi@sicau.edu.cn (Z.H.)

**Keywords:** melatonin, Chinese kale, phenylpropanoids, α-linolenic acid, plant hormone, calcium

## Abstract

Melatonin, a pleiotropic small molecule, is employed in horticultural crops to delay senescence and preserve postharvest quality. In this study, 100 µM melatonin treatment delayed a decline in the color difference index *h** and *a**, maintaining the content of chlorophyll and carotenoids, thereby delaying the yellowing and senescence of Chinese kale. Transcriptome analysis unequivocally validates melatonin’s efficacy in delaying leaf senescence in postharvest Chinese kale stored at 20 °C. Following a three-day storage period, the melatonin treatment group exhibited 1637 differentially expressed genes (DEGs) compared to the control group. DEG analysis elucidated that melatonin-induced antisenescence primarily governs phenylpropanoid biosynthesis, lipid metabolism, plant signal transduction, and calcium signal transduction. Melatonin treatment up-regulated core enzyme genes associated with general phenylpropanoid biosynthesis, flavonoid biosynthesis, and the α-linolenic acid biosynthesis pathway. It influenced the redirection of lignin metabolic flux, suppressed jasmonic acid and abscisic acid signal transduction, and concurrently stimulated auxin signal transduction. Additionally, melatonin treatment down-regulated RBOH expression and up-regulated genes encoding CaM, thereby influencing calcium signal transduction. This study underscores melatonin as a promising approach for delaying leaf senescence and provides insights into the mechanism of melatonin-mediated antisenescence in postharvest Chinese kale.

## 1. Introduction

Chinese kale (*Brassica oleracea* var. *alboglabra*) is a versatile vegetable, featuring both edible leaves and bolting stems, and is extensively cultivated in Southern China and Southeast Asia. Renowned for its delicate flavor and nutritional richness, including carotenoids, ascorbic acid, and glucosinolates, it enjoys popularity among consumers [1,2,3]. Despite its appeal, this green leafy vegetable faces a brief shelf-life, typically lasting only two to three days, characterized by rapid water loss, yellowing, and quality deterioration postharvest [4]. Various treatments, such as 1-methylcyclopropene (1-MCP) [5], light exposure [6], hot water immersion [7], or high-oxygen atmospheric packaging [4], have been explored to prolong senescence and uphold postharvest quality. However, more effective chemical treatments remain elusive.

Melatonin (N-acetyl-5-methoxytryptamine), recognized as both a plant hormone and master regulator, plays a pivotal role in diverse developmental processes and the intricate interplay between plants and their environment [8,9]. Melatonin treatment has exhibited remarkable efficacy in delaying senescence and preserving the quality of horticultural crops during storage and transportation [8]. Firstly, functioning as an antioxidant, melatonin serves as the primary defense against reactive oxygen species (ROS) in plants [10], enhancing the activity of ROS scavenging enzyme genes and, thereby, augmenting the antioxidant capacity of plants. Secondly, melatonin acts as an anti-senescence agent, down-regulating key genes related to leaf senescence and enzymes involved in chlorophyll degradation, constituting one of the earliest observed melatonin-mediated responses [11]. This effect has been reported in *Brassica* vegetables, such as Pak choi [12,13], Chinese flowering cabbage [10], and broccoli [14,15]. Simultaneously, melatonin participates in secondary metabolism, promoting the biosynthesis of glucosinolates and inducing the production of phenylpropanoids like flavonoids and lignin [16]. Throughout postharvest storage, the integrity of the plasma membrane is significantly compromised alongside senescence [17]. Melatonin mitigates the leakage of cell electrolytes, increases phospholipid content, and augments unsaturated fatty acids, thereby preserving the stability of the cell membrane [18]. Finally, it has been substantiated that melatonin regulates the transcription of various elements, including enzymes, receptors, and transcription factors, involved in the biosynthesis and catabolism of plant hormones [8]. Collectively, these findings suggest that melatonin holds promise for postharvest preservation in horticultural crops due to its multifaceted effects. The optimal concentration for postharvest preservation differs among plant species. For example, 1000 μM melatonin treatment was optimal for delaying the lignification of bamboo shoots [19]. Low doses of melatonin (1 μM) were optimal for preserving antioxidants in broccoli florets [16]. Further, 100 μM melatonin was more effective for extending the shelf life of Chinese flowering cabbage [10], sweet cherry [20], and pomegranate [21]. In addition to plant species, the optimal concentrations of melatonin can vary with the stages of fruit development, different treatment conditions (e.g., application method and duration), and storage conditions [22].

While the efficacy of melatonin treatment in postharvest preservation has been demonstrated for various horticultural crops, limited information exists regarding its impact on postharvest Chinese kale. Recently, the advent of transcriptome sequencing (RNA-Seq) technology has introduced an efficient molecular biology research method [23]. This approach has elucidated the molecular mechanisms involved in glucosinolate metabolism under diverse phytohormone treatments and the reconstruction of nutritional architecture influenced by far-red light in Chinese kale [24,25]. In particular, we selected leaf discs for postharvest observations and experiments in this study. The leaf disc is a good system for leaf senescence research, which can unify the test materials and facilitate the observation of yellowing and uniform sampling. This study aims to investigate the influence of melatonin on postharvest Chinese kale and to unravel the mechanisms underlying delayed leaf senescence in Chinese kale during storage.

## 2. Results

### 2.1. Melatonin Reduced Leaf Senescence in Postharvest Chinese Kale

Yellowing serves as a conspicuous indicator of senescence in leafy vegetables. In the control group, leaf discs exhibited minor yellowing and rot in limited areas after three days of storage (Figure 1A). Following five days of storage, leaf discs in the control group displayed varying degrees of yellowing, with a notable increase in the number of rotting discs. In contrast, leaf discs treated with melatonin retained their green hue after three days of storage. After five days, the melatonin-treated group exhibited approximately half the number of yellowing leaf discs compared to the control group, and rotting conditions were visibly alleviated.

The color difference index *h** reflects leaf color, while *a** indicates red (+) or green (−). Throughout storage, both the control and melatonin-treated leaf discs showed declining *h** values and increasing *a** values (Figure 1B,C). However, after five days of storage, melatonin-treated discs displayed an 8% higher *h** value and a 45% lower *a** value compared to the control, aligning with the senescence phenotype.

The total chlorophyll and carotenoid content is consistent with the photos and color difference results. The total chlorophyll content in the control group dropped sharply, and the content in the control at 5 d was only 38.3% of that at 0 d. The content in melatonin-treated leaf discs was 1.2- and 1.3-fold higher than that in the control at 3 and 5 d, respectively. The total carotenoid content of the two groups showed significant differences at 3 d, and the content in the melatonin treatment was 1.1-fold higher than that in the control.

### 2.2. Differential Gene Expression

Comparisons between samples groups C0 vs. C3, C0 vs. C5, C3 vs. C5, C3 vs. M3, and C5 vs. M5 identified a total of 12,654 (6199 up-regulated, 6455 down-regulated), 11,150 (5421 up-regulated, 5729 down-regulated), 6780 (3574 up-regulated, 3206 down-regulated), 1637 (893 up-regulated, 744 down-regulated), and 429 (238 up-regulated, 191 down-regulated) differentially expressed genes (DEGs), respectively (Figure 2A).

### 2.3. Verification of RNA-Seq Data by qRT-PCR

To verify the precision and reproducibility of the RNA-Seq findings, ten DEGs were randomly selected for qRT-PCR analysis. Almost all of these genes exhibited a consistent expression pattern in both techniques, affirming the credibility of the data obtained from RNA-Seq analysis (Appendix A).

### 2.4. Global mRNA Response to Senescence and Melatonin Treatment

Two comparison pairs, C0 vs. C3 and C3 vs. C5, were elected for the analysis of commonly up-regulated and down-regulated genes by Venn diagrams (Figure 2B,C). A total of 1907 DEGs were identified in three control groups, indicating that the continuous up-regulation (981) and down-regulation (926) of these genes are strongly associated with storage and senescence. Kyoto Encyclopedia of Genes and Genomes (KEGG) annotation analysis was conducted on these up-regulated and down-regulated genes to elucidate the potential function of the identified DEGs (Appendix A). As for up-regulated DEGs, most of them were annotated to the KEGG sub-categories “carbohydrate metabolism”, “biosynthesis of other secondary metabolites”, “lipid metabolism”, and “amino acid metabolism” within the main category, “metabolism”; to “plant hormone signal transduction” within the main category, “environmental information processing”; to “plant-pathogen interaction” within the main category, “organismal systems”. When focusing on “metabolism”, the first four pathways annotated to a large number of genes are shown in Figure 2D. “Starch and sucrose metabolism”, “phenylpropanoid biosynthesis”, “glycerophospholipid metabolism”, and “tryptophan metabolism” were enriched, respectively. In down-regulated DEGs, several similar enrichments were observed: the KEGG sub-categories “energy metabolism”, “carbohydrate metabolism”, “lipid metabolism”, and “metabolism of cofactors and vitamins” within the main category, “metabolism”; to “plant hormone signal transduction” within the main category, “environmental information processing”; to “plant-pathogen interaction” within the main category, “organismal systems”. The pathways with DEGs in “metabolism” are “photosynthesis”, “glyoxylate and dicarboxylate metabolism”, “glycerolipid metabolism”, and “porphyrin and chlorophyll metabolism”, respectively (Figure 2E).

The comparison between C3 vs. M3 revealed a higher number of differentially expressed genes (DEGs) (1637) compared to C5 vs. M5 (429), making it the primary focus for subsequent analysis of melatonin’s regulatory impact on gene expression during storage (Figure 2F; Appendix A). Notably, a greater number of genes were up-regulated under melatonin treatment, annotated to “phenylpropanoid biosynthesis”, “glycerophospholipid metabolism”, and “ascorbate and aldarate metabolism” within the main category, “metabolism”, as well as “plant hormone signal transduction” and “plant-pathogen interaction”.

### 2.5. Melatonin Treatment Activated the Phenylpropanoid Biosynthesis in Postharvest Chinese Kale

An examination of phenylpropanoid biosynthesis in Chinese kale during storage and the alterations in the expression patterns of genes in this pathway induced by melatonin were unveiled (Figure 3). In the general phenylpropanoid pathway, the expression of five genes in control samples exhibited a sharp decline after harvest. Melatonin treatment up-regulated phenylalanine ammonia lyase (*PAL*) (gene_Bol037689, gene_Bol005084, and gene_Bol025102) and 4-coumarate-CoA ligase (*4CL*) (gene_Bol026623) by 1.8–3.9-fold. In the flavonoid biosynthesis branch of phenylpropanoid metabolism, most genes were down-regulated in control samples during storage. Similarly, chalcone isomerase (*CHI*) (gene_Bol038972), flavonol synthase (*FLS*) (gene_Bol024486), isoflavone 2′-hydroxylase (*I2′H*) (gene_Bol018620, gene_Bol028922, gene_Bol027250, gene_Bol043644), and vestitone reductase (*VR*) (gene_Bol030178) were notably induced by 1.7-, 1.6-, 1.9–2.7-, and 2.9-fold in the melatonin-treated discs. In the lignin biosynthesis branch, where most genes in control samples were up-regulated during storage, melatonin treatment had varied effects. Melatonin treatment had different effects on them. Hydroxycinnamoyl-CoA shikimate (*HCT*) (newGene_5137, newGene_5136, and gene_Bol045823), caffeoyl shikimate esterase (*CSE*) (gene_Bol020424), cinnamoyl-CoA reductase (*CCR*) (newGene_3749 and gene_Bol030400), and cinnamyl alcohol dehydrogenase (*CAD*) (gene_Bol028365) were up-regulated by 2.2–2.8-, 2.1-, 1.5–3.2-, and 2.1-fold in melatonin-treated samples. In contrast, melatonin treatment down-regulated caffeate/5-hydroxyferulate 3-*O*-methyltransferase (*COMT*) (gene_Bol038837), ferulate 5-hydroxylase (*F5H*) (gene_Bol018672), and coniferyl-aldehyde dehydrogenase (*REF*) (gene_Bol007617) by 67%, 40%, and 36%. Some genes (gene_Bol031505, gene_Bol015835, gene_Bol015833, and gene_Bol020121) of peroxidase (*POD*) were up-regulated (2.7–3.8-fold) by melatonin treatment, and some (newGene_11372, gene_Bol024475, gene_Bol028999, gene_Bol015540, gene_Bol025599) were down-regulated by 35–86%.

### 2.6. Melatonin Treatment Induced the Lipid Metabolism in Postharvest Chinese Kale

The majority of genes involved in lipid metabolism exhibited up-regulation during storage in control samples, with melatonin treatment intensifying this up-regulation (Figure 4A). Among the DEGs related to cutin, suberine, and wax biosynthesis, the melatonin group showed 1.6–4.1- and 2.3-fold higher expression levels of fatty acid omega-hydroxylase (*FAD*) (gene_Bol030226 and gene_Bol027110) and wax-ester synthase (*WSD1*) (gene_Bol039789) compared with the control. In contrast, fatty acyl-CoA reductase (*FAR*) (gene_Bol013804) was down-regulated by 63%. Concerning DEGs linked to glycerophospholipid metabolism, melatonin treatment resulted in 2.7-, 2.2–2.7-, 3.0-, and 1.5-fold higher expression levels of phospholipase A1 (*DAD1*) (gene_Bol041048 and gene_Bol037159), ethanolaminephosphotransferase (*EPT1*) (newGene_8523, newGene_7830, and newGene_7831), choline-phosphate cytidylyltransferase (*PCYT1*) (gene_Bol013293), and phospholipase A2 (*TGL4*) (gene_Bol005479) compared with the control group. Glycerophosphoryl diester phosphodiesterase (*glpQ*) (gene_Bol012590) was decreased by 78%. Regarding DEGs involved in α-linolenic acid metabolism, acyl-CoA oxidase hydroperoxide dehydratase (*AOS*) (gene_Bol035942) and lipoxygenase (*LOX2S*) (gene_Bol034868) exhibited increased expression by 1.6- and 1.7-fold compared with the control group, while alcohol dehydrogenase class-P (*ADH*) (gene_Bol027580) and allene oxide cyclase (*AOC*) (gene_Bol029161) were down-regluated by 49% and 38%. Additionally, in the biosynthesis of unsaturated fatty acids, the expression level of 17beta-estradiol 17-dehydrogenase (*KAR*) (gene_Bol043251) was up-regulated by 1.7-fold in melatonin-treated samples (Figure 4A).

### 2.7. Melatonin Treatment Influenced Other Metabolic Pathways in Postharvest Chinese Kale

In the ascorbate and aldarate metabolism pathway, most genes in the control group were slightly up-regulated during storage, and melatonin treatment further increased their expression levels (Figure 4C). Specifically, in melatonin-treated samples, the expression levels of inositol oxygenase (*MIOX*) (gene_Bol022755 and gene_Bol042279), UDP-sugar pyrophosphorylase (*USP*) (newGene_2510 and newGene_3331), glucuronokinase (*GLCAK2*) (gene_Bol043354), and L-ascorbate oxidase (*AOXL*) (gene_Bol015809 and newGene_7190) were 1.6–2.0-, 1.4–2.2-, 2.0-, and 1.6–4.1-fold higher than those in the control, while L-ascorbate peroxidase (*APX*) (gene_Bol029036) was decreased by 42%.

In the carotenoid biosynthesis pathway, melatonin treatment down-regulated the expression levels of zeaxanthin epoxidase (*ABA1*) (gene_Bol019241 and gene_Bol037233) and 9-cis-epoxycarotenoid dioxygenase (*NCED3*) (gene_Bol035582) by 48–53% and 59%, respectively, compared with the control group. Conversely, xanthoxin dehydrogenase (*ABA2*) (gene_Bol006111) and abscisic acid 8′-hydroxylase (*CYP707A*) (gene_Bol024384, gene_Bol032060, and gene_Bol005615) were up-regulated by 5.7- and 1.6–2.2-fold (Figure 4D).

### 2.8. Melatonin Treatment Induced Plant Hormone Signal Transduction in Postharvest Chinese Kale

A total of 46 enriched DEGs were associated with the plant hormone signal transduction pathway, including auxin (AUX), cytokinin (CTK), gibberellin (GA), abscisic acid (ABA), ethylene (ETH), brassinosteroid (BR), jasmonic acid (JA), and salicylic acid (SA) sub-pathways (Figure 5A,B).

Five genes encoding auxin/indole acetic acid protein (AUX/IAAs) (gene_Bol040867 and gene_Bol038617) and auxin-responsive GH3 (GH3) (gene_Bol013052, gene_Bol027952, and gene_Bol018258) were up-regulated by 1.5–4.9- and 2.4–4.7-fold in the melatonin-treated group, except for two down-regulated small auxin-up RNA (*SAUR*) genes. In the CTK signal pathway, the expression of CTK receptor histidine kinases (*CRE1*) (gene_Bol004697 and gene_Bol004694) and downstream transcription activators A-Arabidopsis response regulators (*A-ARR*) (gene_Bol022049 and gene_Bol032811) were up-regulated by 2.4–2.6- and 2.1–3.5-fold, while two *B-ARR* genes (gene_Bol015120 and gene_Bol020058) exhibited the opposite trend. Considering the GA signal, one gene encoding the gibberellin receptor (GID1) (gene_Bol036237) was up-regulated by melatonin treatment, whereas another genes encoding DELLA protein (DELLA) (gene_Bol004667) were down-regulated. DEGs included three up-regulated and two down-regulated phytochrome-interacting factor (*PIF*) genes.

Concerning the ABA signal, the expression levels of genes encoding the abscisic acid receptor PYL family (PYL) (gene_Bol044030 and gene_Bol025407) and protein phosphatase 2C (*PP2C*) (gene_Bol012794, gene_Bol033793, and gene_Bol038202) were induced by 1.7–1.8- and 1.8–3.6-fold after melatonin treatment, with the exception of one *PYL* gene (gene_Bol009697) and one *PP2C* gene (gene_Bol026318). In the BR signaling pathway, one gene encoding brassinosteroid-insensitive 1-associated receptor kinase 1 (BAK1) (gene_Bol011094) and one gene encoding brassinosteroid-resistant 1 (BZR1) (gene_Bol039912) were down-regulated by 59% and 34%, respectively, after melatonin treatment, while one gene encoding BR-signaling kinase (BSK) (gene_Bol037085) and three genes encoding xyloglucosyl transferase TCH4 (TCH4) (gene_Bol014220, gene_Bol012209, and gene_Bol039563) were up-regulated by 2.4- and 2.3–3.4-fold in melatonin-treated samples. Regarding the JA and SA signaling, three genes encoding jasmonate ZIM domain-containing protein (JAZ) (gene_Bol044840, gene_Bol029321, and gene_Bol043451), one gene encoding transcription factor TGA (TGA) (gene_Bol010308), and one gene encoding pathogenesis-related protein 1 (PR1) (newGene_329) were up-regulated by 1.5–1.8-, 1.6-, and 2.0-fold after melatonin treatment, while one transcription factor *MYC2* (gene_Bol023805) gene was not, decreased by 86%.

### 2.9. Melatonin Treatment Enhanced Calcium Signal Transduction in Postharvest Chinese Kale

The DEGs involved in plant–pathogen interaction were focused on calcium signal transduction (Figure 5C). The results indicated that the expressions of one cyclic nucleotide gated channel (*CNGC*)-related gene (gene_Bol023929), six cam modulin/calmodulin-like protein (*CaM*/*CML*)-related genes (gene_Bol027610, gene_Bol010737, gene_Bol039637, gene_Bol028942, gene_Bol043000 and gene_Bol020573), and three calcium-dependent protein kinase (*CDPK*)-related genes (newGene_11868, gene_Bol015383, and gene_Bol031964) were up-regulated by 3.9-, 1.6–3.5-, and 1.5–5.5-fold in samples treated with melatonin. One gene encoding CNGC (gene_Bol030279), two genes encoding CDPK (gene_Bol017122 and gene_Bol018715), and one gene encoding respiratory burst oxidase (RBOH) (gene_Bol013343) were decreased by 82%, 37–70%, and 69% in samples treated with melatonin.

## 3. Discussion

The role of melatonin in delaying the senescence of postharvest horticultural crops has been confirmed [8]. In this study, the obvious inhibition of yellowing and consistent surface color results were observed after melatonin treatment (Figure 1). Transcriptome analysis confirmed that melatonin-induced antisenescence primarily involved the regulation of phenylpropanoid biosynthesis, lipid metabolism, plant signal transduction, and calcium signal transduction (Figure 6).

The phenylpropanoid metabolism plays a pivotal role in both self-development and environmental interactions [26,27]. PAL facilitates the conversion of phenylalanine to cinnamic acid. Meanwhile, 4CL generates *p*-coumaroyl-CoA in an adenosine triphosphate-dependent manner [26]. In our study, melatonin treatment up-regulated *PAL* and *4CL* expression (Figure 3). Within the flavonoid pathway, CHI transforms chalcones into flavanones like naringenin and liquiritigenin [28]. Subsequent enzymatic reactions, facilitated by FLS, I2′H, VR, and other enzymes, ultimately yield flavonols and isoflavonoids, respectively [26]. Melatonin treatment also up-regulated the expression of these genes. Lignin biosynthesis, another prominent downstream branch of the phenylpropanoid pathway [29,30], contributes to forming a physical barrier and reducing pathogenic toxicity [31]. The gateway enzyme, HCT, was up-regulated by exogenous melatonin, redirecting metabolic flux to flavonoids when silenced [32]. The final steps in monolignol biosynthesis involve CCR and CAD, both up-regulated by melatonin treatment. Intriguingly, melatonin down-regulated genes encoding COMT and F5H by over 40%, potentially due to melatonin’s impact on metabolic flux redirection present in lignin biosynthesis. These are key genes for G-S lignin biosynthesis [33]. Melatonin treatment favored the caffeic acid pathway, forming the precursor of G-lignin. However, the implications of this preference in enhancing the resistance of postharvest Chinese kale and the specific reasons for its occurrence warrant further investigation. Nevertheless, melatonin treatment activated phenylpropanoid biosynthesis, enhancing fungal disease resistance and postharvest quality, aligning with previous research findings [34,35,36].

Lipid metabolism plays a pivotal role in a plant’s response to stress [37]. The wax forms a protective layer on the surface of *Brassicaceae* plants, adapting to changes in the external environment [38]. This layer critically reduces external mechanical damage and resists bacterial and fungal invasion [39]. Melatonin treatment up-regulated the expression of genes involved in fatty acid composition regulation, such as *FAD*, and in the catalysis of wax ester biosynthesis, such as *WSD1* (Figure 4A) [40]. Glycerophospholipid is a crucial substance closely associated with the cell membrane [41]. The results demonstrated that melatonin treatment increased the expression of genes related to phosphoglycerol synthesis, including *TGL4*, *DPP1*, and *EPT1* (Figure 4A). Similarly, unsaturated fatty acids enhance the fluidity of the plasma membrane, maintaining the normal viscosity of protoplasts, which is essential for cell membrane integrity and postharvest quality [42]. α-linolenic acid, closely linked to the biosynthesis of plant antioxidants and JA biosynthesis [43], is regulated by key enzyme genes, including *DAD1* and *FAD*, both up-regulated after melatonin treatment, affecting the synthesis of α-linolenic acid [44]. Using α-linolenic acid as a substrate, a series of enzymatic reactions catalyzed by *LOX*, *AOS*, *AOC* and other enzymes result in the formation of 12-oxophytodienoic acid (12-OPDA). The three rounds of β-oxidation during the formation of JA and methyl jasmonate by 12-OPDA are regulated by *ACOX* as one of the key enzymes [45]. Melatonin treatment increased the expression of *LOX* and *AOS* while decreasing the expression of *AOC* and *ACOX*. We speculate that melatonin blocks the conversion of 12-OPDA to JA, as observed in melatonin-treated maize under drought conditions [44]. Therefore, exogenous melatonin delays cell membrane damage and promotes the repair of the cell membrane by regulating genes related to membrane lipids. Simultaneously, it prevents the accumulation of main metabolites in the JA biosynthesis pathway.

In plant hormone signal transduction, alterations in the JA signal were observed following melatonin treatment. Melatonin up-regulated the expression of *JAZ* and concurrently down-regulated the expression of *MYC2*. JAZ can bind to bHLH TF, such as MYC2, thereby constraining the expression of early JA-responsive genes [46]. This observation provides further evidence that melatonin inhibits senescence by suppressing the activity of JA response genes. Analogous to JA, there was enrichment in the biosynthesis of ABA, the carotenoid downstream metabolism. Melatonin treatment in this study restrained ABA accumulation by down-regulating the expression of the ABA biosynthesis gene *NCED* and up-regulating the transcription of the ABA catabolic gene *CYP707A*. Additionally, melatonin treatment enhanced the inhibition of SnRK2 and ABFs by up-regulating the activities of PYL, the receptor protein of ABA, and PP2Cs. Comparable results were identified in previous studies where melatonin treatment selectively down-regulated *MdNCED3* and up-regulated catabolic genes *MdCYP707As*, thereby reducing ABA accumulation in drought-stressed *Malus* plants [47]. In a separate study, melatonin suppressed ABF-mediated abscisic acid biosynthesis and chlorophyll degradation, consequently delaying leaf senescence in Chinese flowering cabbage [10]. Furthermore, melatonin treatment influenced the signal transduction of other hormones. It positively impacted the expression of AUX signals (*AUX*/*IAA* and *GH3*) and elevated the gene expression level of *A-ARR5* in the cytokinin (CTK) signal, consistent with observations in loquat seedlings [48]. Similarly, after melatonin treatment, the gene encoding GID1 was induced, and the degradation of DELLA may be attributed to the relatively high level of GA binding to GID1. Moreover, melatonin treatment up-regulated the expression of defense genes, such as *TGA* and *PR1,* in the SA signal. Therefore, exogenous melatonin treatment influences plant hormone signal transduction to effectively delay senescence. In this study, we used the transcriptome to preliminarily identify the plant hormones that may play major roles in alleviating aging caused by melatonin. In our further studies, we will focus on plant hormones, including quantifying plant hormones and exploring possible transcriptional regulatory mechanisms, etc.

Calcium serves as an integral participant in various plant responses to both biotic and abiotic stress, functioning as an intracellular second messenger [49]. CNGCs represent a group of non-selective cation channels. CDPK and CaM, two primary types of calcium-sensor proteins, play roles in RBOH-mediated ROS burst and NO production, dependent on Nitric Oxide synthase (NOS) in plants, respectively [50]. In this study, melatonin-mediated senescence inhibition mitigated the ROS burst while augmenting *CaM* expression and seemingly favoring NO production. This contradicts results obtained in cherry tomato fruit, where melatonin acted as a CaM antagonist [34]. This divergence might be attributed to the essential role of ROS burst in melatonin-induced disease resistance. Furthermore, the up-regulation of ascorbate and aldarate metabolism in our study underscores melatonin’s role in maintaining ROS balance, consistent with findings in other studies [16]. Collectively, melatonin treatment may entail distinct resistance mechanisms during senescence and disease.

## 4. Materials and Methods

### 4.1. Plant Materials and Melatonin Treatments

The typical white-flowered ‘Sijicutiao’ Chinese kale (*Brassica oleracea* var. *alboglabra*) was harvested at 30 days of growth from the experimental base of the Horticulture College at Sichuan Agricultural University. The seeds of Chinese kale were sown on 10 May 2022. All plants were grown in culture chambers under 12 h light/12 h dark conditions with 80 μmol m ^−2^·s ^−1^ of luminous intensity at (22 ± 1) °C, with 70% relative humidity. The entire Chinese kale plants were subjected to spraying with either 0 μM (control) or 100 μM melatonin (Solarbio Science and Technology Co., Ltd., Beijing, China) solutions, the latter being the determined optimum dose from a pre-experiment. The daily spraying treatment was conducted in darkness to prevent melatonin decomposition for three days preceding harvest. Plants of consistent size and devoid of mechanical damage were carefully chosen and transported to the laboratory. The third leaf from the bottom of 40 plants in each treatment was collected, cut into 1 cm diameter discs. Six leaf discs were cut from the same leaf and quickly and randomly placed in a petri dish with 0.5% agar. Sixteen leaf discs were placed in each petri dish, with 15 dishes per treatment. The leaf disks in three petri dishes were used for photography and surface color measurement, two dishes were used for pigment content determination, and two dishes were used for RNA extraction in transcriptome sequencing. Following storage for 0, 3, and 5 days, the samples underwent photography and surface color analysis, and they were subsequently frozen in liquid nitrogen and stored at −80 °C for subsequent transcriptome analysis.

### 4.2. Surface Color Measurement

Surface color of leaf discs was measured using a chromameter (NR110, 3nh Co., Ltd., Shenzhen, China). We randomly selected three leaf discs in each petri dish as a repeat and recorded the color. Three petri dishes were used as three repetitions of the surface color, with the average taken as the levels of *h** and *a** [51]. Negative chromaticity (*a**) indicates green, and positive chromaticity (*a**) indicates red. *h** is the hue angle. Hue angle starts at the +*a** axis and is expressed in degrees (e.g., 0° is +*a**, or red, and 180° is −*a**, or green).

### 4.3. Chlorophyll and Carotenoid Content

The sample powder was ground and extracted with acetone, and the supernatant was filtered and analyzed by high-performance liquid chromatography (HPLC). Samples (10 μL) were separated using isopropanol and 80% acetonitrile–water at a flow rate of 0.5 mL min^−1^. Result of chlorophyll and carotenoids content was expressed as mg g^−1^ of fresh weight [51].

### 4.4. Transcriptome Sequencing

Leaf discs from control treatments at 0 days (C0), 3 days (C3), and 5 days (C5), as well as melatonin-treated samples at 3 days (M3) and 5 days (M5), were collected. Biomarker Technologies Co., Ltd. (Beijing, China) conducted transcriptome analysis using three replicates. The total RNA was extracted using the RNAprep Pure Plant Kit (Tiangen, Beijing, China) according to the instructions provided by the manufacturer. A total amount of 1 μg RNA per sample was used as input material for the RNA sample preparations. Sequencing libraries were generated using NEBNext UltraTM RNA Library Prep Kit for Illumina (NEB, Ipswich, MA, USA) following manufacturer’s recommendations, and index codes were added to attribute sequences to each sample. The raw reads were further processed with a bioinformatic pipeline tool, BMKCloud (www.biocloud.net (accessed on 7 April 2023)) online platform. Hisat2.0.4 tools software was used to map with reference genome. Gene expression levels were estimated by fragments per kilobase of transcript per million fragments mapped. Differential expression analysis of two conditions/groups was performed using the DESeq2.1.30.1 [52]. The resulting *p* values were adjusted using Benjamini and Hochberg’s approach for controlling the false-discovery rate. Genes with an adjusted *p*-value < 0.05 found by DESeq2.1.30.1 were assigned as differentially expressed. We used KOBAS3.0 [53] software to test the statistical enrichment of differential expression genes in KEGG pathways. Raw data underwent processing to eliminate adaptors and low-quality reads. High-quality paired-end clean reads were aligned to the Brassica oleracea reference genome (http://brassicadb.org/brad/datasets/pub/Genomes/Brassica_oleracea/V1.1/ (accessed on 7 April 2023)) using HISAT2.2.0.4 (Appendix A). Differential expression analysis employed the DESeq2_edgeR program, with DEGs defined based on screening criteria: fold change ≥ 1.5 and *p*-value < 0.05. Subsequently, all DEGs were primarily annotated using the KEGG database (https://www.kegg.jp/kegg/ (accessed on 7 April 2023)).

### 4.5. qRT-PCR Validation

RT-qPCR was performed following the instructions of the TB Green Premix Ex Taq II (Tli RNaseH Plus, Takara, Tokyo, Japan) kit on a Bio-Rad iCycler thermocycler (Bio-Rad, Hercules, CA, USA). Relative gene expression levels were calculated using the formula 2^−ΔΔCT^. Three replicates were conducted in qRT-PCR. The primers used in this study are listed in Appendix A.

### 4.6. Statistical Analysis

Data were analyzed using one-way ANOVAs. Histogram and heatmap were generated using Origin 2023 (OriginLab Corporation, Northampton, MA, USA).

## 5. Conclusions

The transcriptome data generated in this study substantiated that melatonin effectively delays leaf senescence in postharvest Chinese kale. Melatonin treatment induced an up-regulation in the expression of pivotal enzyme genes within the general phenylpropanoid pathway, exemplified by *PAL*, concurrently stimulating the expression of genes involved in flavonoid and lignin synthesis. Likewise, genes responsible for wax and unsaturated fatty acid synthesis in the lipid metabolism, namely *WSD*, *DAD*, and *FAD*, exhibited an up-regulation. Conversely, melatonin treatment led to the inhibition of biosynthesis pathways, such as JA and ABA, along with the downstream metabolism of α-linolenic acid and carotenoid. Notably, JA and ABA signal transduction were also suppressed, while AUX signal transduction was promoted. Additionally, adjustments to the calcium signaling pathway were made to impede the burst of ROS. In summary, this study unveiled the intricate mechanism by which melatonin delays leaf senescence in postharvest Chinese kale.

## Figures and Tables

**Figure 1 ijms-25-02250-f001:**
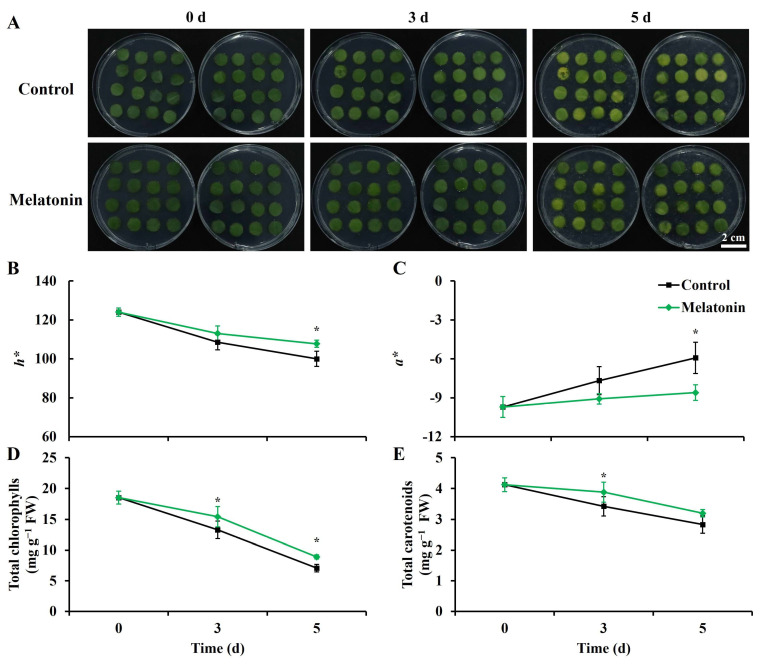
Effect of melatonin treatment on visual appearance and color-related indexes in Chinese kale leaf discs during storage. (**A**) Visual appearance of Chinese kale leaf discs at each sampling time. Scale bar = 2 cm. (**B**,**C**) Effect of melatonin treatment on color parameters (*h** and *a**) in Chinese kale discs during storage. (**D**,**E**) Effect of melatonin treatment on total chlorophylls and carotenoids content in Chinese kale discs during storage. Asterisks (*) indicate the significant differences (*p* < 0.05) between control and melatonin-treated samples.

**Figure 2 ijms-25-02250-f002:**
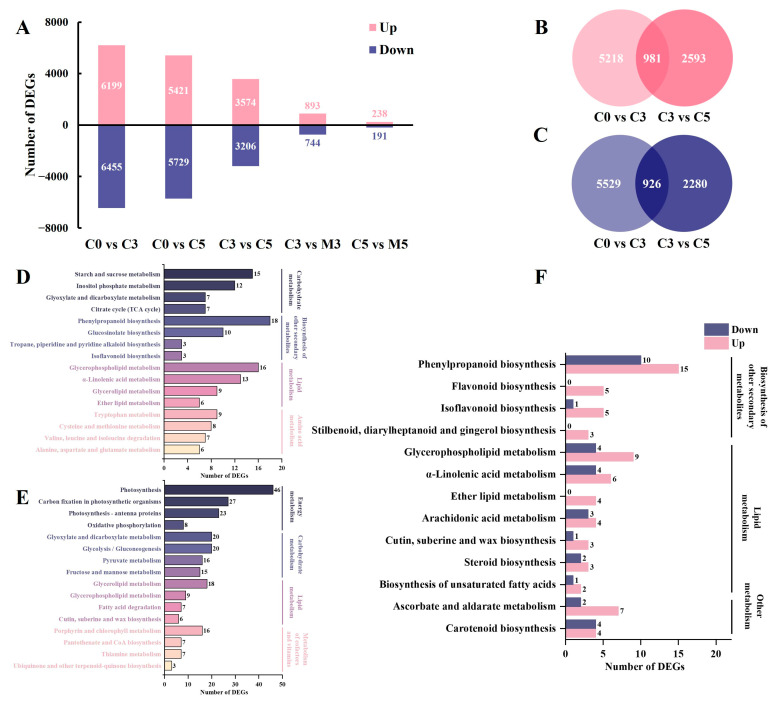
The distribution characteristics of differentially expressed genes (DEGs) in postharvest Chinese kale discs treated with melatonin or not. (**A**) The number of DEGs between comparison pairs. (**B**) Venn diagram illustrating up-regulated DEGs commonly and uniquely expressed among the control groups during storage. (**C**) Venn diagram illustrating down-regulated DEGs commonly and uniquely expressed among the control groups during storage. (**D**) Kyoto Encyclopedia of Genes and Genomes (KEGG) classification annotated by commonly up-regulated DEGs among the control groups during storage. (**E**) KEGG classification annotated by commonly down-regulated DEGs among the control groups during storage. (**F**) KEGG classification annotated by DEGs of the comparison pair C3 vs. M3. The numbers in D, E and F represent the number of DEGs enriched in KEGG pathway.

**Figure 3 ijms-25-02250-f003:**
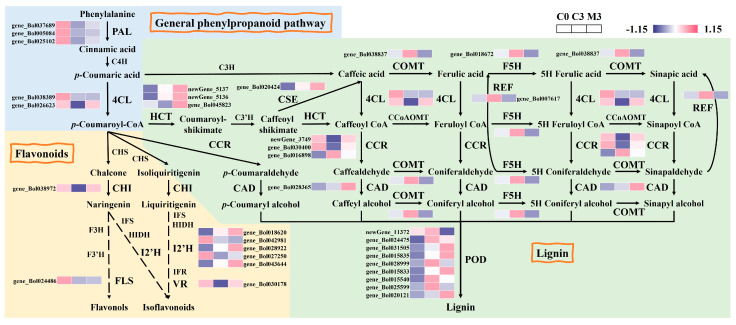
Heatmap of differentially expressed genes in phenylpropanoid biosynthesis in postharvest Chinese kale discs treated with melatonin or not.

**Figure 4 ijms-25-02250-f004:**
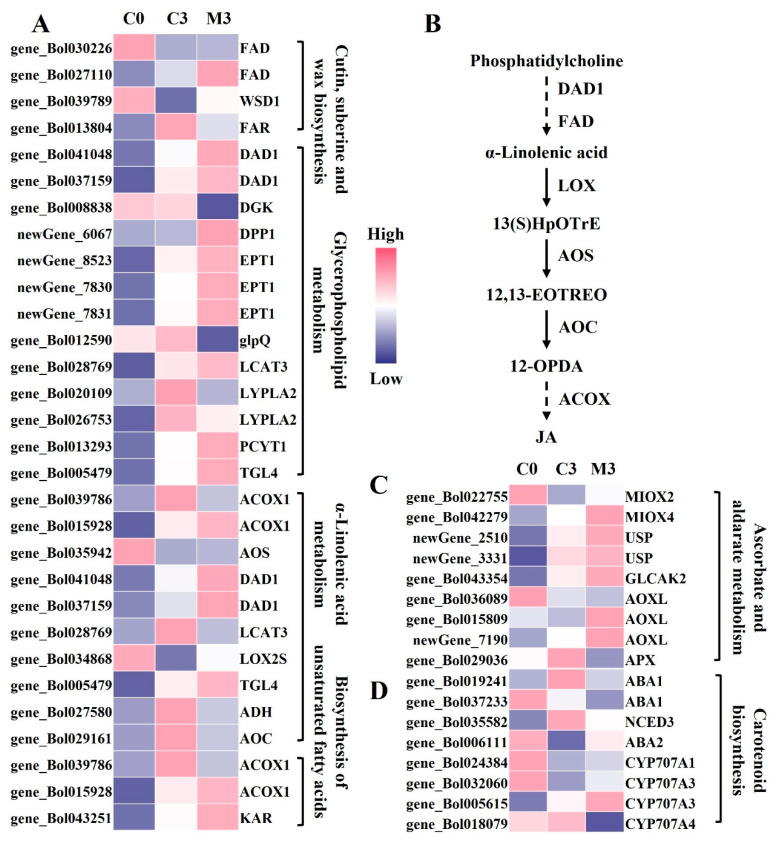
Heatmap of differentially expressed genes (DEGs) and metabolism pathway in postharvest Chinese kale discs treated with melatonin or not. (**A**) Heatmap of DEGs in lipid metabolism. (**B**) α-linolenic acid metabolism pathway. (**C**) Heatmap of DEGs in ascorbate and aldarate metabolism. (**D**) Heatmap of DEGs in carotenoid biosynthesis.

**Figure 5 ijms-25-02250-f005:**
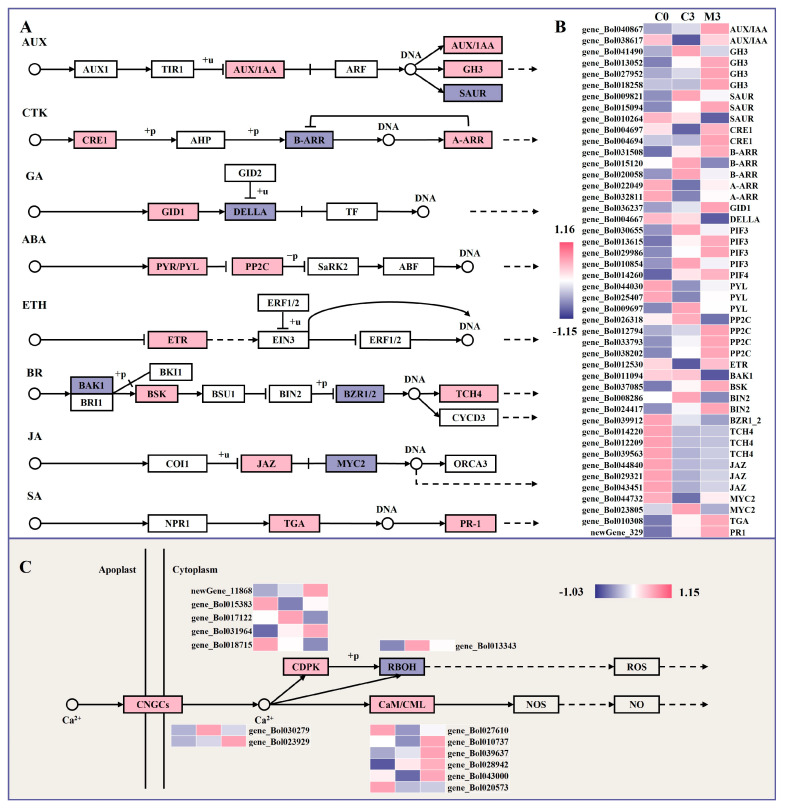
Heatmap of differentially expressed genes (DEGs) and metabolism pathway in postharvest Chinese kale discs treated with melatonin or not. (**A**,**B**) Heatmap of DEGs in plant hormone signal transduction and metabolism pathway. (**C**) Heatmap of DEGs in calcium signal transduction and metabolism pathway. The proteins marked in pink box indicates up-regulated genes, the proteins marked in purple box indicates down-regulated genes and white represents no significant change.

**Figure 6 ijms-25-02250-f006:**
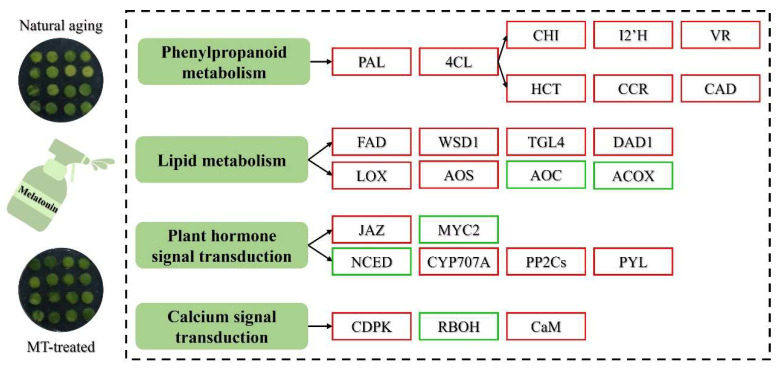
Model of the regulatory mechanism through which melatonin treatment delays the senescence of Chinese kale. The red color represents gene up-regulation, and the green color represents gene down-regulation.

## Data Availability

All data supporting the findings of this study are available within the paper and within its Appendix A published online.

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
