# Peer review of "Transcriptome Analysis Reveals the Mechanism by Which Exogenous Melatonin Treatment Delays Leaf Senescence of Postharvest Chinese Kale (Brassica oleracea var. alboglabra)"

_ijms, 2024, doi:10.3390/ijms25042250_

Round 1
Reviewer 1 Report
Comments and Suggestions for Authors
The authors have studied the transcriptomic response of Chinese Kale by exogenous application of melatonin. They found that melatonin can delay leaf senescence in Chinese kale. In China, Kale is a extensively cultivated vegetable, featuring both edible leaves and bolting stems. Transcriptome analysis revealed 1637 differentially expressed genes after 3 days of storage, implicating its influence on phenylpropanoid biosynthesis, lipid metabolism, and plant signal transduction pathways. Authors highlight melatonin as a promising tool for extending shelf life in Chinese kale by unraveling its multifaceted antisenescence mechanisms.
I don’t have many comments about this study, but I would be happy if authors can explain;
1. Why are observations of the phenotype at these time points (1 day, 2 days, and 4 days) not reported after melatonin treatment?
2. Why did you choose genes randomly to verify by qPCR instead of those potentially involved in senescence? For example biosynthesis genes of phytohormones….
3. I would suggest quantifying the Phytohormone content if possible.
Regarding Discussion, It should be concise.
It would be great if you can explain your methodology of RNA extraction, library preparation and detailed analysis etc. Also please cite the reference if possible for NR110 chromameter (link etc) . how did you measure Surface color measurement?
Figure 1. caption and description should be detail and it should stand alone. Same for other figures.
Comments on the Quality of English Language
may be minor editing of English language required
Reviewer 2 Report
Comments and Suggestions for Authors
Main comments:
The authors of the manuscript undertook an interesting research topic. Melatonin is the subject of extensive scientific research, and its effect on plants is also studied. The aging process of plants during natural development and after harvest, in the green state, is determined by many physiological factors in the plant. This process depends on the cultivation conditions in which the plant grew, including soil and climate, and the time of harvest, which certainly had a large impact on the physiological condition of the plant for storage. The aim of the research undertaken by the authors of the manuscript was to investigate the effect of melatonin on Chinese kale after harvest and to discover the mechanisms underlying the delayed aging of Chinese kale leaves during storage. The first part of the research for the adopted purpose was carried out very sparsely.
The most important issue is the method of conducting the experiment - the authors conducted very partial and perhaps even non-representative research on small leaf sections, where there is no normal aging of entire leaf blades, but dehydration of small leaf fragments. Testing the chlorophyll content and other physiological parameters would be very useful in assessing the effect of melatonin. However, colorimeter-only testing itself seems to be very limited in this scientific experiment and does not seem to be the best way to assess leaf aging. The descriptions in the "Materials and Methods" chapter also seem to be insufficient.
Detailed comments and suggestions:
Abstract:
The concentration of melatonin used in the research should be provided. No mention was made of studies using a colorimeter.
Introduction
There should also be information in this part of the manuscript, among others: what concentrations of melatonin were applied to plants by other researchers, in what publications?
Line 37: There should also be the full name of the chemical (1-MCP)
Line 68-69: what exactly does it mean “the reconstruction of nutritional architecture”.
Results
Line 75 -76: what does control group and all control group mean? Were the plants, for example, sprayed with distilled water?
Line 88: Figure 1, the word “phenotype” should not be used here. A sparse description of the photos and a caption below them.
Materials and Methods
Line 363: Cultivar 'Sijicutiao', what time of year was it grown and in what conditions?
Line 366: What was sprayed with, what was the control, spraying with distilled water? Where was melatonin obtained from, manufacturer?
Line 367: Based on what research did the authors determine the most appropriate concentration of melatonin. Would a higher concentration be harmful and a lower concentration less effective?
Line 371: One disc is cut from one leaf? Where were these cut discs of leaves placed, how many pieces, and where were they stored? How many colorimeter measurements were taken in each treatment?
Line 373. What observations?
Line 376: Should be: Colorimeter or chromameter (NR110…)
Line 378: Explain what h* and a* mean
Comments on the Quality of English LanguageMini Minor stylistic errors in sentence structure.
Round 2
Reviewer 1 Report
Comments and Suggestions for Authors
Thank you for considering my suggestions.
Author Response
Thanks.
Reviewer 2 Report
Comments and Suggestions for Authors
The authors significantly improved the manuscript and added omissions and additions indicated by the reviewer. It still requires improvement according to the indications and suggestions below:
Abstract:
Line 16-17: In my opinion, the inserted sentence is not completely correct, I suggest you think again and take into account the accuracy. What pigment was tested? how were the photos assessed (color difference) and what were the measurements used to measure? The phrase "according to the photos" is not the best one here.
Line 108: Figure 1. You certainly can't call it "morphological indices", this term refers to other features.
Line 423: Please enter in units like in Figure 1, not “g kg−1”.
